# Lung Cancer Targeted Chemoradiotherapy via Dual-Stimuli Responsive Biodegradable Core-Shell Nanoparticles

**DOI:** 10.3390/pharmaceutics14081525

**Published:** 2022-07-22

**Authors:** Roshni Iyer, Harish Ramachandramoorthy, Trinh Nguyen, Cancan Xu, Huikang Fu, Tanviben Kotadia, Benjamin Chen, Yi Hong, Debabrata Saha, Kytai Truong Nguyen

**Affiliations:** 1Department of Bioengineering, University of Texas at Arlington, Arlington, TX 76019, USA; rosh25iyer@gmail.com (R.I.); harish.ramachandramoor@mavs.uta.edu (H.R.); txn1130@mavs.uta.edu (T.N.); cancan.xu@mavs.uta.edu (C.X.); huikang.fu@mavs.uta.edu (H.F.); yihong@uta.edu (Y.H.); 2Joint Bioengineering Program, University of Texas Southwestern Medical Center, Dallas, TX 75390, USA; 3Department of Biology, University of Texas at Arlington, Arlington, TX 76019, USA; tanviben.kotadia@mavs.uta.edu; 4Department of Radiation Oncology, University of Texas Southwestern Medical Center, Dallas, TX 75390, USA; benjamin.chen@utsouthwestern.edu

**Keywords:** nanoparticles, stimuli-responsive, targeted drug delivery, chemoradiotherapy

## Abstract

Lung cancer is one of the major causes of cancer-related deaths worldwide, primarily because of the limitations of conventional clinical therapies such as chemotherapy and radiation therapy. Side effects associated with these treatments have made it essential for new modalities, such as tumor targeting nanoparticles that can provide cancer specific therapies. In this research, we have developed novel dual-stimuli nanoparticles (E-DSNPs), comprised of two parts; (1) Core: responsive to glutathione as stimuli and encapsulating Cisplatin (a chemo-drug), and (2) Shell: responsive to irradiation as stimuli and containing NU7441 (a radiation sensitizer). The targeting moieties on these nanoparticles are Ephrin transmembrane receptors A2 (EphA2) that are highly expressed on the surfaces of lung cancer cells. These nanoparticles were then evaluated for their enhanced targeting and therapeutic efficiency against lung cancer cell lines. E-DSNPs displayed very high uptake by lung cancer cells compared to healthy lung epithelial cells. These nanoparticles also demonstrated a triggered release of both drugs against respective stimuli and a subsequent reduction in in vitro cancer cell survival fraction compared to free drugs of equivalent concentration (survival fraction of about 0.019 and 0.19, respectively). Thus, these nanoparticles could potentially pave the path to targeted cancer therapy, while overcoming the side effects of conventional clinical therapies.

## 1. Introduction

Lung cancer has been considered one of the deadliest forms of cancer with a 5-year survival rate of less than 15% [1,2]. Surgical removal of non-small cell lung cancer (NSCLC) also poses a problem as many patients have unresectable tumors [3,4]. Individual chemotherapy and radiation therapy are the conventional treatments for lung cancer. However, these strategies face several limitations that affect their therapeutic efficacy, including rapid relapse, poor drug bioavailability and off-target side-effects, which severely affect the drug dosage used to achieve the maximal therapeutic efficacy [5]. Cancer cells are also inherently resistant to chemotherapeutic drugs, resulting in minimal therapeutic efficacy and eventually leading to the expansion and proliferation of cancer cells and tumor growth [6]. For instance, Cisplatin, an FDA approved chemotherapeutic drug for the treatment of NSCLC, which crosslinks with purine bases in DNA and causes DNA damage and subsequently initiates cell apoptosis [7], is extremely toxic with side-effects such as nephrotoxicity, neurotoxicity and ototoxicity, in addition to being severely limited in its therapeutic actions due to cancer cell resistance [8,9]. Additionally, the DNA self-repair mechanisms of cancer cell DNA severely limit the benefits of radiation therapy, often resulting in poor overall survival of lung cancer patients [5].

Chemoradiotherapy (CRT) can overcome these limitations by enhancing the effectiveness of radiation therapy. CRT is the concurrent administration of a chemotherapeutic agent and radiation therapy to control tumor growth, and subsequently improve patient responses to cancer therapy [10]. The most commonly investigated drug for combined treatment with radiation is Cisplatin, due to its effectiveness in patients with widespread NSCLCs [11]. CRT combining Cisplatin and radiation has been investigated before in the clinic to study its effectiveness to treat NSCLC, and was observed to significantly improve the five-year survival rate of patients undergoing CRT treatment compared to sequential chemo-radiotherapy (where the patient is administered the chemotherapeutic drug, followed by radiation therapy after a few chemotherapy drug dosing regimens) and/or radiation therapy alone [12,13].

However, a major disadvantage of CRT is the increased risk of side-effects such as anemia, acute esophagitis and neutropenia, thus limiting the drug dosages that can be implemented without causing toxicity to the patients, while maintaining the drug within the therapeutic window [13]. The systemic administration of Cisplatin is also limited by severe toxicity, especially when combined with radiation. Chemotherapy also has a great patient to patient variability in treatment efficiency. Similarly, other major chemotherapeutic drugs such as Carboplatin, Paclitaxel and Docetaxel have also been administered in significantly lower doses when combined with radiation [11]. Furthermore, the addition of a chemotherapy regimen to a radiation therapy regimen may not be sufficient to achieve the desired therapeutic efficacy. Thus, there is an increasing need to develop strategies that can achieve adequate therapeutic efficacy, while reducing the toxicity associated with the combined treatment.

Recent developments in multi-compartment nanoparticles (NPs) for cancer therapy and their unique characteristics such as controlled drug releases of various drugs for combined therapies and targeting capabilities have made them suitable carriers for CRT. Multi-compartment NPs or core-shell NPs are composed of two or more materials that are combined to form a core and shell of the NPs [5,14,15]. Varieties of core-shell materials have been investigated for drug delivery applications, such as metal based NPs coated with either a polymeric or silica shell, multi-shell NPs, hollow shell NPs and organic core-shell NPs among others [16]. For example, multifunctional core-shell NPs for lung cancer chemo-radiotherapy were developed for the treatment of lung cancer [5]. These NPs were comprised of a folic acid-functionalized poly (N-isopropyl acrylamide)/carboxymethyl chitosan shell encapsulating gemcitabine hydrochloride (Gem), coated onto a PLGA core pre-loaded with NU7441. These NPs exhibited dual temperature and pH-responsive Gem release kinetics from the shell and sustained NU7441 release kinetics from the core. Additionally, these NPs observed an enhanced therapeutic efficacy when combined with radiation therapy, compared to radiation alone.

In this project, we developed dual-drug loaded, dual-stimuli responsive core-shell nanoparticles (DSNPs) that can deliver chemotherapeutic agents and radiosensitizers synergistically, for combined chemotherapy and radiation therapy to treat lung cancers (Figure 1). In this research, we investigated the combination of NU7441+Cisplatin, a CRT drug regimen that has been investigated before [5]. NU7441 is a radiosensitizer that inhibits the DNA-PKs repair pathway in cancer cells, therein inducing apoptosis [17,18,19].NU7441 works synergistically with radiation treatment increasing its efficiency. We utilized a GSH-sensitive polymer (PU-SS) synthesized by our lab to form the core NP, which encapsulates and delivers the chemotherapeutic drug, Cisplatin [20]. Due to the presence of reducible disulfide linkages in the backbone of the polymer, the elevated levels of GSH in cancer cells trigger the release of Cisplatin from the NPs [20,21]. These NPs were coated with a radiation-responsive shell made of hyaluronic acid. Hyaluronic acid (HA) is a natural polymer, composed of glyosidic bonds in its backbone that degrade in the presence of reactive oxygen species (ROS) and free radicals generated during radiation therapy [22]. The shell encapsulates the radiosensitizer, NU7441, which would be released from the shell in response to radiation exposure, to synergistically enhance the therapeutic effects of radiation therapy. Much research has recently been investigated on Hyaluronic acid-based nanoparticles for targeted delivery for cancer treatment [23,24,25,26]. It has also been shown that NPs made from HA or HA-conjugates were highly bio-compatible. Wang et al. investigates a co-polymer of Hyaluronic acid NPs that has a targeting effect on cancer cells with higher circulation and tumor retention [23]. To improve the lung cancer cell targeting capabilities of these NPs, we functionalized these NPs with antibodies against EphA2 receptors (E-DSNPs) that are highly expressed on the surface of lung cancer cells, while being poorly expressed on healthy lung cells [27,28,29]. In this paper, we will discuss the synthesis and use of these E-DSNPs to improve internalization to lung cancer cells and subsequent therapeutic efficacies.

## 2. Materials and Methods

### 2.1. Materials

Polyvinyl alcohol (PVA, MW 15,000–25,000), N-hydroxysuccinimide (NHS), bovine serum albumin (BSA), Dulbecco’s Modified Eagle’s Medium (DMEM), hexafluoroisopropanol (HFIP), EDC (1-ethyl-3-(3-dimethylaminopropyl)-carbodiimide) and Triton R X-100 were purchased from Sigma-Aldrich (St. Louis, MO, USA), and dichloromethane from Merck (Kenilworth, NJ, USA). Cisplatin was ordered from Cayman Chemicals (Ann Arbor, MI, USA). Hyaluronic acid was purchased from Lifecore Biomedical (Chaska, MN, USA). NU7441 was obtained from Selleckchem Chemicals (Houston, TX, USA). Fetal bovine serum (FBS), 1X trypsin EDTA and penicillin-streptomycin were ordered from Invitrogen (Waltham, MA, USA). All reagents were of analytical grade.

### 2.2. Cell Lines and Culture Conditions

Alveolar Type 1 (AT1) cells were purchased through Applied Biological Materials Inc. (Richmond, BC, Canada). EphA2 positive tumor lines A549 were purchased through American Type Culture Collection (ATCC, Manassas, VA, USA). A549 cell line was cultured in DMEM medium supplemented with 10% heat inactivated FBS (Corning, NY, USA), 100 U/mL penicillin, and 100 µg/mL streptomycin (Invitrogen, Waltham, MA, USA). AT1 cells were cultured in IMDM medium supplemented with 10% heat inactivated FBS, 100 U/mL penicillin.

### 2.3. Synthesis of DSNPs

Core-shell NPs comprised of a Glutathione-responsive nanoparticle (GNP) core and Hyaluronic acid nanoparticle (HNP) shell were synthesized via an electrostatic layer-by-layer coating technique, where the negatively charged GNPs are coated with an intermediate positively charged layer, followed by coating with a negatively charged HA, thus forming a sandwich of alternating negative and positive charged layers (Figure 1). A biodegradable polyurethane comprised of disulfide linkages in the backbone (PU-SS) was synthesized from polycaprolactone diol (molecular weight = 2000), hexadiisocyanate and hydroxyethyl disulfide with a molar ratio of 0.2:2:1.8 [20,21]. Cisplatin loaded GNPs were synthesized using a previously established protocol [21]. Briefly, Cisplatin (10% *w/w*) was dissolved in 200 µL of DMSO. The solution was added dropwise to the solvent (5% HFIP in DCM) containing PU-SS (2% *w/v*). The solution was sonicated for 5 min at 10 watts. It was then added to 4 mL of PVA (5% *w/v*) followed by ultrasonication for 5 min at 30 watts. The solution was stirred overnight for solvent evaporation, centrifuged, washed, and collected at 15,000 RPM, 30 min. The nanoparticles were lyophilized with Trehalose (5% *w/w*) as cryoprotectants. About 10 mg of these NPs were dispersed in 10 mL of 10 mM sodium chloride (NaCl) solution and vortexed to thoroughly suspend the NPs. 250 µL of polyallylamine hydrochloride (PAH, 1000 µg/mL in 10 mM NaCl; MilliporeSigma, Burlington, MA, USA) was added to this suspension to coat a positive charge on the GNPs and allowed to react for 30 min at room temperature while stirring, following which the NP suspension was centrifuged at 15,000 RPM for 30 min to collect the PAH coated GNPs. The pelleted NPs were suspended in 10 mL of deionized water, and 1 mL of hyaluronic acid (10 mg/mL in deionized water) containing 2 mg of NU7441 was added to the NP suspension to initiate the coating of the drug loaded HA onto the GNP cores. The solution was allowed to stir overnight at room temperature, followed by centrifugation at 15,000 RPM for 30 min and lyophilized with Trehalose (5% *w/w*) as cryoprotectants for 2 days to collect the DSNPs.

Antibody (Ab) conjugation onto the DSNPs was performed via carbodiimide crosslinker chemistry. Briefly, 5 mg of lyophilized NPs were suspended in 4 mL of 2% 2-ethanesulfonic acid (MES; MilliporeSigma) buffer (~pH 5.0), and 20 mg of 1-Ethyl-3-(3-dimethylaminopropyl) carbodiimide (EDC; MilliporeSigma) was added to this suspension and allowed to react at room temperature for 30 min. 20 mg of N-hydroxysuccinimide (NHS; MilliporeSigma) was then added to this suspension and allowed to react at room temperature for another 30 min (this reaction activates ester groups on the surface of the NPs for conjugating of ligands/antibodies onto NPs). About 11 µg of Abs was then added to the above NP suspension, and the conjugation reaction was performed overnight at 4 °C, followed by dialysis to remove unconjugated Abs, and lyophilization with Trehalose (5% *w/w*) as cryoprotectant to obtain the Ab/targeting ligand-conjugated NPs or E-DSNPs. Antibody conjugation efficiency was determined by Bradford protein assays and calculated by the following equation:$$Ab\;coating\;efficiency=\frac{Amount\;of\;Ab\;used-Amount\;of\;Ab\;in\;supernatant\;}{Amount\;of\;Ab\;used}\times 100\%$$

### 2.4. Characterization of Nanoparticles

Particle size was measured using the Dynamic Light Scattering (DLS) technique via the ZetaPALS zeta potential analyzer (Brookhaven Instruments Inc., Holtsville, NY, USA). NU7441 loading efficiency into the shell and Cisplatin loading efficiency into the core of the DSNPs were determined by an indirect method. Briefly, the amounts of NU7441 and Cisplatin in the supernatants were determined by a UV-Vis spectrophotometer (Tecan, Morrisville, NC, USA) at an absorbance of 288 nm and 346 nm, respectively.

The drug loading efficiency was calculated by the following:$$Drug\;loading\;efficacy=\frac{Drug\;amount\;used-Drug\;in\;supernatant}{Drug\;amount\;used}\times 100\%$$

To study the drug release kinetics, NPs were exposed to either of the 4 conditions, no treatment (PBS), GSH (5 mM), radiation (5 Gy) and a combination of GSH (5 mM) and radiation (5 Gy). The purpose of this study was to observe the synergistic Cisplatin release (from the GSH sensitive core) and NU7441 release (from the radiation responsive shell). To initiate the drug release, 1 mg/mL of DSNPs were resuspended in either PBS (0 mM GSH) or PBS containing 5 mM of GSH (6 replicates each). Three replicates from each GSH concentration were then exposed to radiation (0 Gy or 5 Gy). The NPs were then incubated at 37 °C between predetermined time points. At each time point, the NP suspensions were centrifuged at 15,000 RPM for 25 min. The supernatants containing the released drugs were collected and stored at −20 °C, following which the NPs were resuspended in PBS (±5 mM GSH) and incubated until the next time point. The amounts of NU7441 released were calculated using a standard curve of NU7441 (absorbance: 234 nm), followed by normalizing the drug released against the amount of NU7441 loaded into the NPs. Similarly, Cisplatin was calculated using a standard curve of Cisplatin (absorbance: 310 nm), followed by normalizing the drug released against the amount of Cisplatin loaded into the NPs. A sample size of *n* = 3 per group was used for this study.

### 2.5. Investigating the Therapeutic Efficacies of Dual Drug Combinations

The therapeutic efficacy of dual drug combinations compared to single drug administrations were determined by studying the A549 cancer cell killing ability of the drugs via MTS assays (72 h of drug exposure) and colony formation assays (10 days of drug exposure). To study the drug therapeutic efficacy via MTS assays, A549 lung cancer cells were seeded at a seeding density of 20,000 cells/well in a 48 micro-well plate and allowed to attach overnight. The following day, cells were treated with free NU7441, free Cisplatin (0.5 µg/mL and 1 µg/mL drug concentration) or a combination of the drugs (NU7441+Cisplatin). Untreated cells and cells treated with 1% Triton X-100 were regarded as the negative and positive controls, respectively. For this study, we used a sample size of *n* = 4 per treatment group. 72 h later, cell viability was determined by MTS assays (Promega Corporation, Madison, WI, USA) following the manufacturer’s protocol.

The therapeutic efficiency of the drugs was further analyzed using the Colony formation assay (CFA). CFA studies were performed using a previously established protocol [30]. Briefly, A549 lung cancer cells were seeded in a 60 mm petri dish. The petri dish was treated with either free drug or the drug combination (100 ng/dish). The dish was incubated for 10 days at 37 °C. The petri dishes were washed with PBS and stained with crystal violet (MilliporeSigma) dye (0.5% *w/v* in 6% *v/v* glutaraldehyde). The colonies with at least 50 cells in each dish were counted. The study was performed with *n* = 3 dishes per treatment group at each time point.

### 2.6. Cytotoxicity Analysis of Nanoparticles

To analyze the cytocompatibility of the NPs, human lung alveolar Type 1 epithelial cells (AT1 cells) were seeded in a 96 well plate at 8000 cells/well. After cell confluence, they were treated with non-drug loaded E-DSNPs at concentrations of 0, 50, 100, 200, 500, and 1000 µg/mL. The cells were then incubated at 37 °C for 24 h, and the cell viability was determined using MTS cell viability assays (CellTiter 96^®^AQueous One Solution Cell Proliferation Assay, Promega Corporation, Madison, WI, USA).

### 2.7. Hemocompatibility Analysis of Nanoparticles

The hemocompatibility evaluation of the non-drug loaded E-DSNPs were analyzed by hemolysis study. Briefly, human blood was acquired from a donor followed by methods approved by the Institutional Review Board at the University of Texas at Arlington. NPs were analyzed by incubating non-drug loaded E-DSNPs in activated blood (0.1 M calcium chloride) at NP concentrations of 100 and 1000 µg/mL and compared with water and 0.9% saline as controls for 2 h at 37 °C. The samples were centrifuged at 1000× *g*, and the supernatant was analyzed using UV-Vis spectroscopy at 545 nm. The percentage of hemolysis was calculated using the following equation:$$Hemolysis=\frac{Abs\;of\;sample-Abs\;of\;negative\;control}{Abs\;of\;positive\;control-Abs\;of\;negative\;control}\times 100\%$$

The blood clotting kinetics of the NPs were analyzed by incubating non-drug loaded E-DSNPs in activated blood (0.1 M calcium chloride) at NP concentrations of 100 and 1000 µg/mL and compared with 0.9% saline as control. At pre-determined time points of 10, 20, 30, and 60 min, water was added to the tubes to lyse the red blood cells (RBCs) that were not part of the formed clot. The blood clotting kinetics were observed visually, and the supernatant was analyzed using UV-Vis spectroscopy at 540 nm absorbance wavelength. Both hemolysis and blood clotting studies were performed at a sample size of *n* = 8 per group.

### 2.8. Cellular Uptake of Nanoparticles

Cellular uptake of DSNPs and E-DSNPs was determined by measuring the amount of fluorescently labeled NPs that become internalized by lung cancer cells. DSNPs and E-DSNPs loaded with a fluorescent dye, Coumarin-6 (MilliporeSigma), were synthesized as described above by using Coumarin-6 dye instead of the Cisplatin drug inside the GSH core. Coumarin-6 was used to facilitate fluorescence mediated detection of NPs in the cells. To study in vitro cellular uptake of these NPs, A549 human lung cancer cells were seeded at a seeding density of 10,000 cells/well in a 96 micro-well plate and allowed to attach overnight. The next day the cells were incubated with DSNPs or E-DSNPs at different concentrations (0, 25, 50, 100, 250 and 500 µg/mL with *n* = 4 per concentration per group) at 37 °C for 2 h. Post 2 h, cells were washed thrice with sterile PBS. The cells were lysed using Triton X-100 (1% *v/v*). The number of NPs internalized was determined by measuring the Coumarin-6 fluorescence intensity at wavelength of λex = 458 nm and λem = 540 nm. The intensity translated as µg of NPs was normalized against the total protein content from each well determined using bicinchonic acid (BCA) assays following the company’s instructions (Pierce™ BCA Protein Assay Kit, ThermoFisher Scientific, Waltham, MA, USA). Furthermore, fluorescence imaging was used to image E-DSNP and DSNP uptake into A549 lung cancer cells. Briefly, A549 lung cancer cells were seeded at a cell seeding density of 150,000 cells on glass coverslips and allowed for overnight attachment. The following day, the cells were exposed to 1 mg/mL of DSNPs or E-DSNPs encapsulating Coumarin-6 for 2 h. The cells were washed three times with PBS and fixed for 15 min in 4% paraformaldehyde. The fixed cells were washed with PBS to remove excess paraformaldehyde and their nucleus stained with NucBlue^®^ (ThermoFisher Scientific, Waltham, MA, USA) dye. The cells were then observed under a fluorescence microscope (Cytoviva Inc., Auburn, AL, USA).

### 2.9. In Vitro Therapeutic Efficacy

A study to determine the benefits of using dual drugs vs. single drugs was first performed using MTS assays. The in vitro therapeutic efficacy of dual drug-loaded E-DSNPs compared to single drug-loaded DSNPs in the absence of radiation was determined. The purpose of this study was to first determine the enhanced therapeutic efficacy of dual drug loaded NPs against free drug and single drug loaded NPs. The drug combination was chosen from the results obtained from the drug therapeutic efficacy study. Briefly, A549 and H460 lung cancer cells were seeded at a seeding density of 20,000 cells/well in a 48 micro-well plate and allowed to attach overnight. The following day, cells were treated with either the free single drugs (0.5 or 1 µg/mL drug concentration), or combined free drugs, or drug-loaded NPs, i.e., E-DSNPs with NU7441, E-DSNPs with Cisplatin, or E-DSNPs with NU7441+Cisplatin. Untreated cells and cells treated with 1% Triton X-100 were regarded as the negative and positive controls, respectively. 72 h later, cell death was determined by MTS cell viability assays. In addition, the toxicity of the DSNPs vs. E-DSNPs (without radiation) to healthy lung cells (AT1) was also studied by assessing cell viability of the AT1 cells via MTS cell viability assays. For this study, the cells were treated with 500 µg/mL of drug loaded NPs for 2 h, after which the media was aspirated, cells were washed 2X with media to remove leftover NPs and replenished with fresh media (this was accomplished to prevent any excess non-specific uptake of the NPs).

The in vitro therapeutic efficacies of dual-drug loaded E-DSNPs compared to single drug loaded E-DSNPs in the presence of radiation were determined by studying the viability of A549 lung cancer cells after treatment by MTS assays. The study was performed as described above, this time with exposure to 5 Gy radiation in addition to E-DSNPs. The radiation source used in this study was 137Cs gamma irradiator with a current dose rate of 2.86 Gy/min (Energy 661.7 KeV, JL Shepherd, Model Mark 1-68, San Fernando, CA, USA). 72 h after exposure, cell viability was determined by MTS assays. To observe the survival of cancer cells upon exposure to the drugs, we also performed colony formation assays (CFA). To perform these studies, A549 lung cancer cells were seeded on 60 mm petri dishes, and in vitro CFA studies were performed as described previously [30]. The cells were treated with free NU7441+Cisplatin (100 ng/mL of each drug/dish), DSNPs (loaded with NU7441+Cisplatin), E-DSNPs (loaded with NU7441+Cisplatin) and blank nanoparticles, following which they were irradiated (3 sets, each treated at either 2, 5, or 10 Gy) and incubated at 37 °C for a period of 10 days. Another set of replicates was also generated, without exposure to radiation (0 Gy). This set served as a control for the study. The petri dishes were washed with PBS and stained with crystal violet (MilliporeSigma) dye (0.5% *w/v* in 6% *v/v* glutaraldehyde). The number of colonies with at least 50 cells/colony in each dish were then counted, and survival fraction of the cells was calculated and plotted. To calculate the survival fraction of A549 cells, the number of surviving colonies at a radiation dose (S_D_) was normalized (S_D_/S_0_) to the number of surviving colonies at 0 Gy radiation (S_0_). The data was fit with the Universal Survival Curve [31] with the implementation of the linear quadratic model at doses below a transition dose, and for doses larger than a transition dose, the multitarget model was incorporated.

### 2.10. Statistical Analysis

Results were analyzed statistically using one-way ANOVA and Tukey’s multiple comparisons test analysis (GraphPad Prism 9.2.0 software, San Diego, CA, USA) with *p* < 0.05 considered as a significant value. All results were displayed as mean ± SD, and quadruplet samples (*n* = 4) were used for each experiment if not specified.

## 3. Results

### 3.1. Physico-Chemical Characteristics of DSNPs

The GSH NP core (GNP), non-drug loaded DSNPs and dual-drug loaded DSNPs observed a diameter of 187 ± 28 nm, 223 ± 31 nm and 297 ± 27 nm respectively, by dynamic light scattering technique (Figure 2A). All NPs had a polydispersity index (PDI) of ~0.3, confirming the homogeneity of the NPs. Conjugation of antibodies against EphA2 further increased the NP size to 323 ± 21 nm. The Zeta potential change of NPs in different stages of NP synthesis shows the layer-by-layer coating and high stability of the DSNPs (Figure 2B). Scanning electron microscopy (SEM) imaging of the DSNPs observed a spherical morphology of the NPs (Figure 2C).

NU7441 and Cisplatin loading efficiencies into the E-DSNPs were measured indirectly via a UV-vis spectrophotometer at an absorbance of 288 nm and 346 nm, and was calculated to be about 71% and 56%, respectively. Drug release of Cisplatin and NU7441 from the E-DSNPs under GSH (5 mM) and radiation (5 Gy) exhibited glutathione-responsive and radiation-responsive drug characteristics (Figure 2C,D). E-DSNPs exposed to PBS alone produced negligible NU7441 and Cisplatin release. Similarly, E-DSNPs exposed to GSH alone also observed poor Cisplatin and NU7441 release. The poor Cisplatin release could potentially be due to the HA layer acting as a shield to prevent drug leakage from the GSH sensitive core. Radiation alone, on the other hand, triggered the degradation of the HA shell, characterized by the significant increase in NU7441 (~90%) and Cisplatin (~40%) within 72 h. Furthermore, NPs treated with GSH and radiation experienced nearly 90% Cisplatin and NU7441 release within 72 h, as a result of radiation degrading the radiation-responsive shell; thus, exposing the GSH sensitive core to GSH in the solution.

### 3.2. In Vitro Cancer Cell Killing Efficacy of Drug Combinations

The in vitro therapeutic efficacies of drug combinations were investigated using MTS cell viability and colony formation assays. MTS assay was used to study the effect of combined NU7441 and Cisplatin, over a 72-h period (Figure 3A). The combination of free NU7441 and Cisplatin significantly reduced the viability of A549 cells to 40%, compared to NU7441 and Cisplatin alone (cell viability reduced to 60% and 55%, respectively). We also examined the efficacy of a combined drug regimen vs. individual drug therapy by evaluating their ability to reduce survival of A549 lung cancer cell colonies via colony formation assays. The results observed a significant decrease in the survival fraction (SF) post treatment with combinations of radiosensitizers and chemotherapeutic drugs compared to the single drugs alone (Figure 3B). The combination of NU7441 and Cisplatin was significantly killing more cancer cells and reducing more cancer cell colony formation (SF: 0.49) than NU7441 alone (SF: 0.90) or Cisplatin alone (SF: 0.67).

### 3.3. Cytocompatibility and Hemocompatibility of E-DSNPs

Hemocompatibility of the E-DSNPs was analyzed by investigating the hemolysis of blood and clotting profiles of whole blood exposed to the NPs. E-DSNPs exhibited less than 5% hemolysis, which is below the acceptable range by the FDA. An increase in hemolysis was observed with an increase in NP concentration. E-DSNPs exhibited a hemolysis of about 0.2% and 1% at a NP concentration of 0.1 mg/mL and 1 mg/mL, respectively. Additionally, the blood clotting trend was similar to that of the control group (0.9% saline). These results confirm the hemocompatibility of the E-DSNPs (Figure 4A).

Cytotoxicity of E-DSNPs at increasing concentrations was determined by MTS cell viability assays. The E-DSNPs observed over 85% cyto-compatibility at the highest NP concentration of 1000 µg/mL to AT1, confirming the cyto-compatibility of the NPs to healthy lung cells (Figure 4B).

### 3.4. Investigation of E-DSNP Uptake into Lung Cancer Cells

Spectrophotometric analysis of NP uptake into A549 cells found that DSNPs and E-DSNPs consisted of NP dose-dependent uptake into A549 lung cancer cells (Figure 5B). Importantly, E-DSNPs significantly improved NP uptake, with nearly a 50% increase in NP uptake at the highest NP concentration. Similar findings were observed using fluorescence microscope, where EphA2 targeted NPs exhibited a distinct difference in NP localization (in terms of higher green intensity as a result of FITC in the NPs) in the A549 cells (Figure 5A).

### 3.5. In Vitro Cancer Cell Killing Efficacy of NPs

In vitro therapeutic efficacy of E-DSNPs in the absence of radiation was investigated in A549 and H460 lung cancer cells to observe the benefits of dual drug therapy compared to that of a single drug (Figure 6). The E-DSNPs loaded with NU7441 and Cisplatin significantly reduced the viability of H460 cells (20% cell viability), out-performing the therapeutic efficacy of either E-DSNPs encapsulating NU7441 (32% cell viability) or E-DSNPs encapsulating Cisplatin (cell viability 34%). The dual drug-loaded E-DSNPs also observed significant improvement in H460 cell death compared to free combined drugs (NU7441+Cisplatin) (cell viability: ~20% and 35%, respectively). A similar trend in therapeutic efficacy was observed for A549 lung cancer cells treated with dual-drug loaded E-DSNPs compared to either E-DSNPs encapsulating NU7441 only or E-DSNPs loaded with Cisplatin only (cell viability: ~30%, 50% and 40%, respectively). The dual drug-loaded E-DSNPs also observed significant improvement in A549 cell death compared to free combined drugs (NU7441+Cisplatin) (cell viability: ~29% vs. 41%).

In vitro therapeutic efficacy of dual-drug loaded E-DSNPs in combination with radiation exposure, was investigated in A549 lung cancer cells, to observe the benefits of dual drug therapy compared to that of a single drug (Figure 7A). The dual drug loaded E-DSNPs significantly reduced the viability of A549 cells (~61% cell viability), out-performing the therapeutic efficacy of E-DSNPs encapsulating NU7441 alone (~84% cell viability) or E-DSNPs loaded with Cisplatin alone (~71% cell viability). The therapeutic efficacy of dual drug-loaded E-DSNPs was further improved by combining the NPs with radiation (5 Gy), where the cell viability was significantly reduced (~29%) compared to radiation alone (~80% viability), free dual drugs (~51% viability) and E-DSNPs encapsulating either NU7441 only (~46% viability) or Cisplatin only (~41% viability). Importantly, the efficacy of combining Cisplatin and NU7441 with concurrent radiation treatment was clearly observed.

The therapeutic efficacy and associated toxicities of NU7441 and Cisplatin loaded E-DSNPs vs. DSNPs to A549 lung cancer cells and AT1 lung epithelial cells were also determined by MTS cell viability assays (Figure 7B). A549 cells treated with dual drug-loaded E-DSNPs and dual drug-loaded DSNPs observed reduced cell survival, although E-DSNPs significantly enhanced A549 cell death (~18% cell viability) compared to DSNPs (~50%) and free NU7441+Cisplatin (~53%). However, in the case of AT1 cells, free NU7441+Cisplatin, DSNPs and E-DSNPs each reduced cell viability to about 50%, potentially due to the poor binding of E-DSNPs to AT1 cells, thus producing similar cell death as the untagged DSNPs.

The cancer cell killing capability of E-DSNPs was also tested by colony formation assays. Photomicrographs (Figure 8A) represent cancer cell colonies from various treatment groups exposed to increasing doses of radiation exposure. The surviving fraction of A549 cells treated with dual drug loaded E-DSNPs concurrently with radiation observed a radiation-dose dependent response, where the survival fraction of the cells was reduced to 0.39, 0.17, and 0.01 (nearly 100% loss of cancer cell survival) when exposed to 2 Gy, 5 Gy and 10 Gy, respectively (Figure 8B). Cells treated with free drugs noted a survival fraction of 0.41, 0.39, and 0.18 upon exposure to 2 Gy, 5 Gy and 10 Gy, respectively. Untargeted DSNPs consisted of a survival fraction of 0.55, 0.30, and 0.14 upon exposure to 2 Gy, 5 Gy and 10 Gy, respectively. Thus, concurrent CRT with our E-DSNPs as drug carriers could significantly improve lung tumor cell killing.

## 4. Discussion

We have reported innovative dual-stimuli responsive core-shell NPs that are comprised of a glutathione-responsive core encapsulating a chemotherapeutic drug and a radiation-sensitive shell loaded with a radiosensitizer, for multiple drug delivery for CRT. These NPs have several exciting features, namely their stimuli-responsiveness that permits enhanced release of a radiosensitizer during concurrent radiation therapy, followed by Cisplatin release in the intracellular environment. Furthermore, the incorporation of EphA2 provided targeting of the NPs to the lung cancer cells, while sparing the healthy cells, thus reducing the chances of off-target drug toxicity. These NPs were loaded with radiosensitizer NU7441 and chemotherapeutic drug Cisplatin for enhanced chemo-radiotherapy of lung cancer and functionalized with EphA2 antibodies to impart lung cancer cell targeting functionalities to the NPs. Our results demonstrate the retention of the stimuli responsive nature of the polymers in the E-DSNPs, where the GSH responsive core observed significant increase in drug release in GSH solutions, while the radiation responsive shell observed significant increase in release of NU7441 with the increase in exposed radiation dose. Additionally, we observed an increased Cisplatin and NU7441 release from the core and shell, respectively when exposed to radiation and glutathione. Importantly, dual drug loaded E-DSNPs and DSNPs combined with radiation significantly enhanced lung cancer cell death compared to NPs encapsulating single drugs or free single or dual-drugs (with or without radiation), thus emphasizing the benefits of: (1) synergistic radiation therapy concurrent with dual drug therapy, and (2) utilizing NPs responsive to dual stimuli for delivering multiple drugs.

Due to the over-expression of EphA2 receptors on cancer cells, we functionalized the DSNPs with antibodies against EphA2 receptors to improve the localization and enrichment of the NPs in the tumor regions. Ephrin transmembrane receptor A2 is a member of the Eph family, which is the largest family of tyrosine kinase receptors [32]. EphA2, highly expressed on lung cancer cells, plays a major role in cancer recurrence and metastasis and often results in poor prognosis and survival of NSCLC patients [27]. Compared to the folate conjugate core-shell NPs developed earlier by our group [5], we observed nearly 25 times higher NP internalization into A549 cancer cells when conjugating antibodies against EphA2 onto the NP surfaces. Similarly, EphA2 targeting also outperformed CD44 and folate targeting by hyaluronic acid micelles synthesized by Liu et al. [33]. As reported by this study, the addition of folate groups to hyaluronic acid micelles did not improve the uptake of the micelles, but conjugation of EphA2 to the hyaluronic acid backbone resulted in a two-fold increase in NP uptake. These results once again highlight the superior and highly specific targeting capability of EphA2. We also prove the effectiveness of our targeting in vitro where the uptake of conjugated nanoparticles was significantly higher than that of un-conjugated nanoparticles by about two-fold.

Stimuli-responsive nanoparticles have been developed and used for cancer therapies effectively. In our study, the synergistic radiation and glutathione responsiveness of the shell and core of the NPs were confirmed. The HA shell acts as a shield to drug leakage from the GSH core, as evidenced by the lack of Cisplatin release in the absence of radiation. NU7441 release from the shell demonstrated a sharp burst release when treated with radiation. The GSH responsiveness of the core is similar to previous reports of redox responsive NPs containing disulfide linkages in the polymer backbone, where the addition of glutathione enhances drug release from the NPs [34]. Radiation induced release of NU7441 from the shell also follows similar reports of ROS induced drug release from hyaluronic acid-based NPs [35]. Chiang et al. [36] reported a single nanocarrier responsive to ROS and GSH levels in the environment. Nearly 100% of campothecin was released from these NPs within 50 h when exposed to either 100 µM of H_2_O_2_ or 20 mM of GSH; however, drug release in response to both stimuli together was not studied. Some NP designs incorporating multiple drugs into a single structure for dual-drug release have also been investigated. Zhang et al. [37] utilized poly(lactic-co-glycolic acid)-poly (ethylene glycol) (PLGA-PEG) NPs co-delivering Cisplatin and Wortmannin (wtmn) to enhance the efficacy of radiation therapy, while reducing side-effects. However, these NPs observed poor drug encapsulation efficiency of about 12.5% and 40% for wtmn and Cisplatin, respectively, potentially due to the two drugs being loaded into the same NPs resulting in one drug interfering with the loading of the other drug. The E-DSNPs in our study, on the other hand, are two-compartment nanoparticles, thus loading of one drug does not affect the loading of the other, as evidenced by the higher drug encapsulation efficiencies of Cisplatin (~56%) and NU7441 (~71%).

Various NP formulations for chemo-radiotherapy (CRT) have been investigated with the central purpose of enhancing therapeutic efficacy, while reducing toxicity to the healthy tissues. Core-shell nanoparticle structures have gained popularity due to their multi-compartment design that permits the loading of multiple drugs and synergistic drug release. Kim et al. [38] fabricated pluronic-based core-shell NPs encapsulating Doxorubicin (dox) in the shell for chemotherapy and gold NPs in the core for radiosensitization of SSC-7 oral cancer cells. The combination of 5 Gy radiation and the Dox loaded NPs observed a fold-reduction in the in vivo tumor volume compared to mice treated with the NPs alone without radiation. However, the combined treatment did not result in a complete tumor ablation, with tumor volumes exhibiting a slightly increasing trend with time. Although in vitro therapeutic efficacies of these NPs combined with radiation was not investigated, the NPs decreased the viability of SSC-7 cells in a Dox dose-dependent manner in this study. In addition, the Dox loaded NPs were not as effective as free Dox at killing the cancer cells (cancer cell viability of ~50% and 30% after 2 days of treatment with the NPs and free Dox, respectively). E-DSNPs, on the other hand, produced a more distinct cancer cell killing efficacy, with nearly a fold-reduction in cancer cell viability compared to the free drugs, even in the absence of radiation. A single compartment design delivering Cisplatin and wtmn lowered the survival of ovarian cancer cells in a radiation dose-dependent manner, with significant increases in sensitization enhancement ratios (SER) from 1.00 to 1.29 for untreated cells and cells treated with dual drug loaded NPs, respectively [37]. The core-shell E-DSNPs also observed significant loss in cell viability (up to ~60%) compared to untreated cells without any radiation exposure. Furthermore, radiation treatment reduced the cell viability to 18% (i.e., ~82% cell death), potentially due to the addition of targeting antibodies against EphA2 transmembrane receptors that enhanced the preferential localization of the NPs.

The application of NPs for CRT has revolutionized pre-clinical cancer treatment, due to the ability to amplify the benefits of both radiation and chemotherapy. Lung cancer chemotherapy using the Campothecin loaded ROS/GSH responsive NPs by Chiang et al. [36] observed a drug concentration dependent decrease in A549 lung cancer cell viability to about 60% within 48 h. Similarly, our dual drug loaded E-DSNPs also decreased A549 cell viability to nearly 30%. Furthermore, E-DSNPs encapsulating NU7441 only reduced the cell viability by 50%, while E-DSNPs encapsulating Cisplatin only reduced the cell viability to 40%. Particularly, exposure to radiation produced a one-fold decrease in cell viability compared to cells treated with E-DSNPs without radiation exposure. Similarly, PLGA NPs encapsulating NU7441 and conjugated with R11 to specifically target and treat PC3 prostate cancer cells, revealed a survival fraction of about 0.5 after radiation exposure, compared to about 0.7 for non-drug loaded R11-PLGA NPs [39]. On the other hand, E-DSNPs combined with radiation reduced the survival fraction of A549 lung cancer cells to less than 0.008, compared to 0.41 by E-DSNPs without radiation treatment. These results further signify the advantages of dual drug treatment from core-shell NPs and synergistic radiation therapy for reducing the survival of cancer cells.

## 5. Conclusions

We synthesized dual drug loaded, dual stimuli responsive NPs, that exhibited both radiation-responsive release of NU7441 from the shell, and glutathione-responsive release of Cisplatin from the core of NPs for concurrent chemo-radiotherapy. These NPs are cyto-compatible and hemocompatible, in addition to exhibiting enhanced uptake because of the EphA2 targeting moieties on the NP surfaces. We also observed enhancement in the therapeutic efficacy of dual drug loaded DSNPs compared to those with a single drug, thus emphasizing the improved therapeutic characteristics of these NPs. Furthermore, the therapeutic efficacy of E-DSNPs in conjunction with concurrent radiation supports the benefits of utilizing E-DSNPs for lung cancer chemo-radiation therapy.

## Figures and Tables

**Figure 1 pharmaceutics-14-01525-f001:**
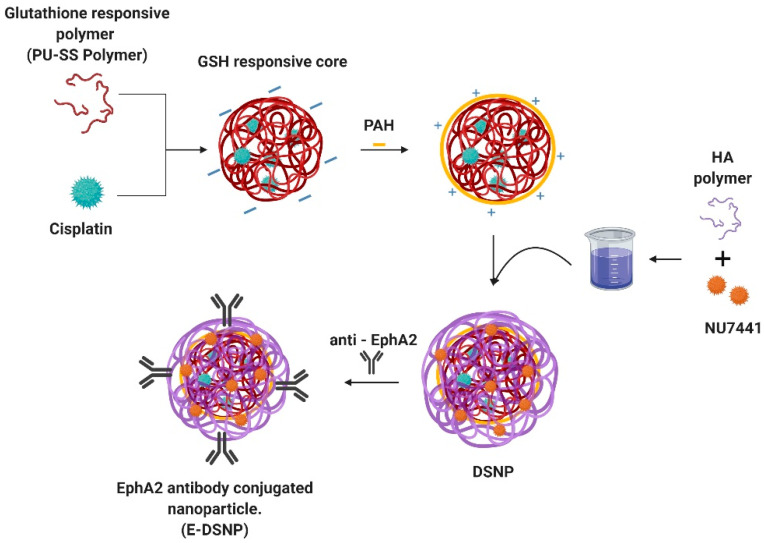
Schematic of core-shell E-DSNPs synthesis by an electrostatic layer-by-layer coating technique.

**Figure 2 pharmaceutics-14-01525-f002:**
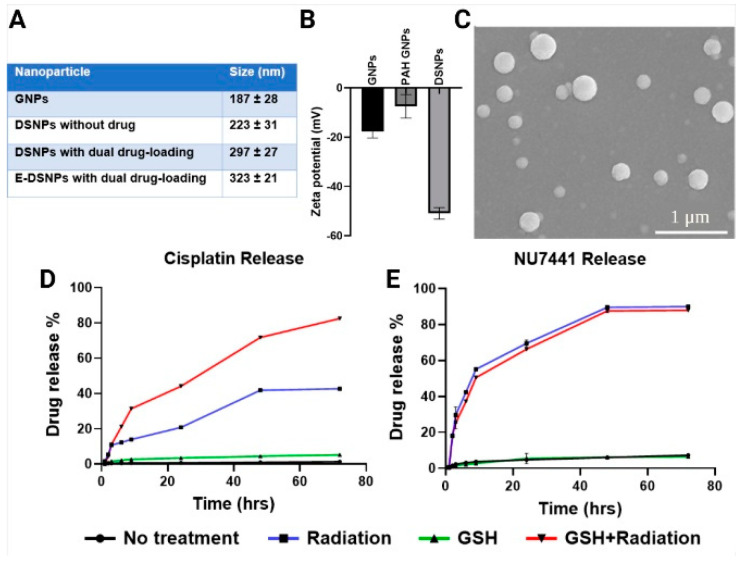
In vitro characterization of E-DSNPs. (**A**) Dual drug-loaded E-DSNPs exhibited hydrodynamic diameter up to ~300 nm. (**B**) Zeta Potential change of the NPs during different layers. (**C**) SEM imaging observed a spherical morphology of the NPs. (**D**) Exposure of E-DSNPs to combined radiation and GSH permitted elevated Cisplatin release from the core of the NPs, compared to that by radiation alone. (**E**) E-DSNPs exhibited higher NU7441 release from the shell, when exposed to combined GSH and radiation than by radiation alone. Exposure to GSH alone did not have any effect on the drug release characteristics of Cisplatin or NU7441.

**Figure 3 pharmaceutics-14-01525-f003:**
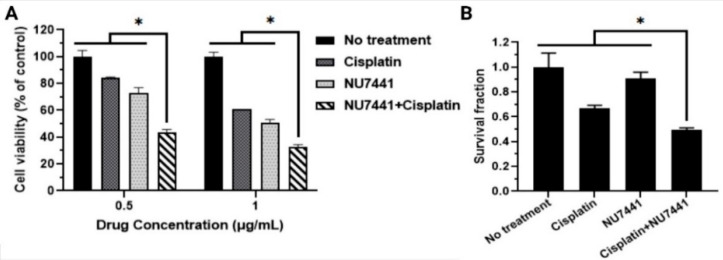
In vitro investigation of dual-drug treatment. (**A**) A549 cell viability via MTS assays after 72 h treatment with drug combinations observed significant decline compared to free drugs. (**B**) Treatment of A549 lung cancer cells with individual free drug vs. combined free drugs revealed lower cancer colony survival in the CFA study as a result of the combined treatment than that with the individual drugs. (* Statistically significant with *p* < 0.05).

**Figure 4 pharmaceutics-14-01525-f004:**
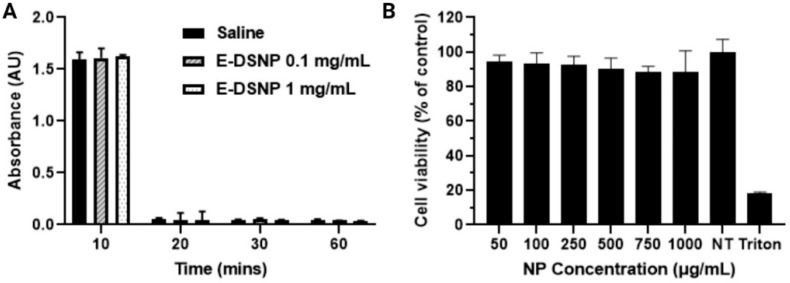
In vitro characterization of E-DSNPs. (**A**) E-DSNPs observed a blood clotting trend similar to that of the saline control. (**B**) In vitro cyto-compatibility of the NPs observed over 90% alveolar type-I cell viability via MTS assays.

**Figure 5 pharmaceutics-14-01525-f005:**
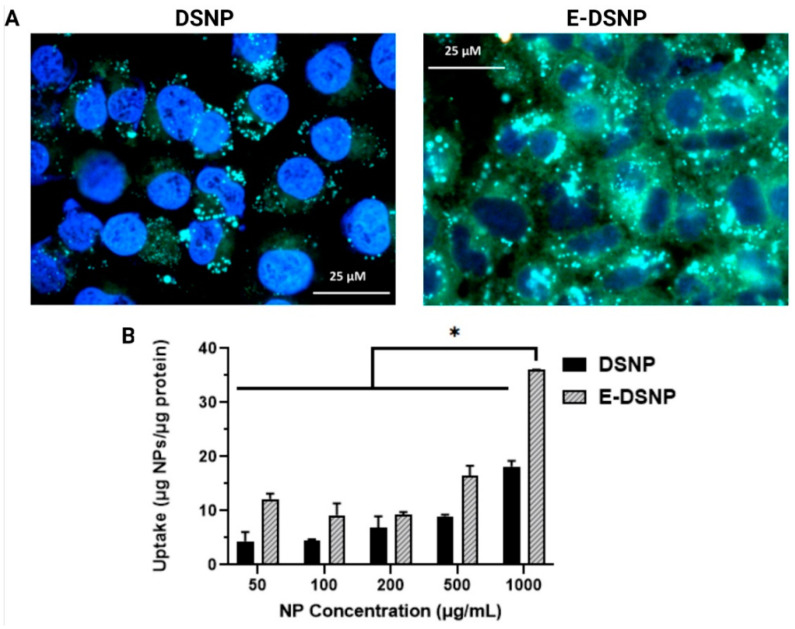
In vitro uptake of E-DSNPs. (**A**) Fluorescence imaging also observed significant enhancement in NP internalization of the anti-EphA2 tagged DSNPs or E-DSNPs (**right**) into A549 cancer cells compared to that of the untargeted DSNPs (**left**). Nuclei of the cells were stained blue with nucBlue^®^ dye. (**B**) Spectrophotometric analysis of E-DSNPs uptake into A549 cells showed dose-dependent uptake, in addition to nearly a fold-increase in NP uptake compared to untargeted DSNPs. (* Statistically significant with *p* < 0.05).

**Figure 6 pharmaceutics-14-01525-f006:**
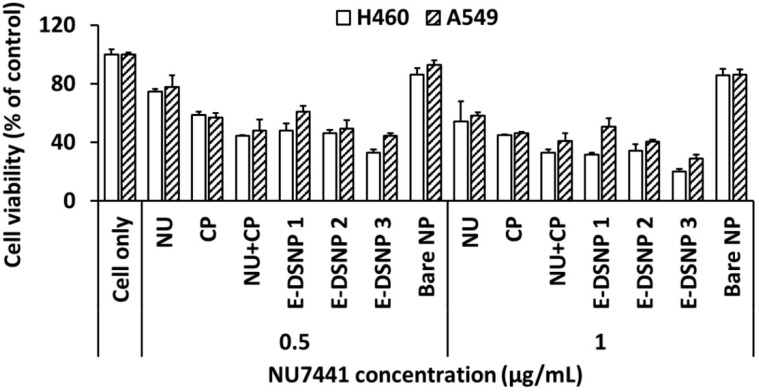
In vitro cancer cell killing efficacies of E-DSNPs. Dual drug loaded DSNPs observed significantly lower cell viability compared to DSNPs encapsulating NU7441 only or Cisplatin only (DSNP1: DSNP with NU7441 only; DSNP 2: DSNP with Cisplatin only, and DSNP 3: DSNP with NU7441+Cisplatin). NU: NU7441 and CP: Cisplatin.

**Figure 7 pharmaceutics-14-01525-f007:**
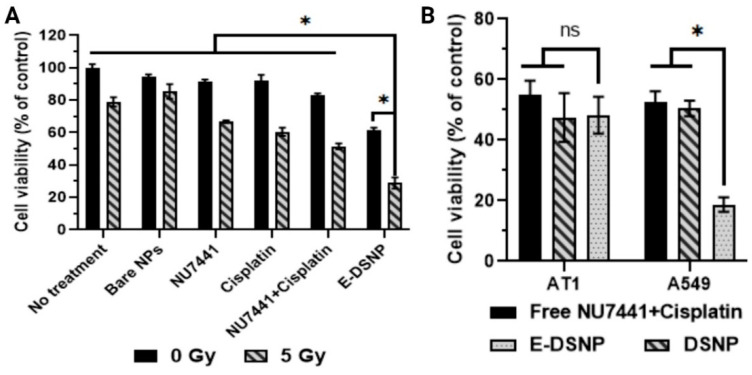
Investigation of in vitro cancer cell killing ability of dual drug loaded E-DSNPs. (**A**) In vitro therapeutic efficacy of dual drug loaded E-DSNPs with 5 Gy radiation treatment observed significantly lower cell viability of A549 lung cancer cells compared to cells treated with E-DSNPs without radiation treatment and single drug loaded NPs or free drug controls (±RT). (**B**) E-DSNPs exert higher toxicity to A549 cells due to targeting via EphA2 receptors but produce similar cancer cell death as untagged controls in AT1 cells due to poor targeting efficiency. (* Statistically significant with *p* < 0.05).

**Figure 8 pharmaceutics-14-01525-f008:**
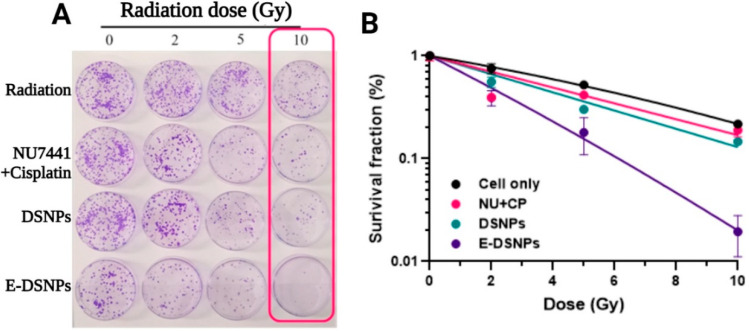
Colony formation assays to study the therapeutic effects of E-DSNPs. (**A**) Photomicrographs of A549 lung cancer cell colonies. (**B**) Graphs depicting survival fractions of cancer cells treated with concurrent 10 Gy radiation and dual drug loaded E-DSNPs treatment to nearly complete loss of cancer cell survival compared to that by radiation, free drugs, and dual drug loaded untargeted NPs. NU: NU7441 and CP: Cisplatin.

## Data Availability

All original data presented in this study will be available on request.

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
