# Peer review of "Lung Cancer Targeted Chemoradiotherapy via Dual-Stimuli Responsive Biodegradable Core-Shell Nanoparticles"

_pharmaceutics, 2022, doi:10.3390/pharmaceutics14081525_

Round 1

Reviewer 1 Report

     This manuscript prepared a layer by layer NPs (DSNP) that are redox and irradiation sensitive. It targets the EphA2 overexpression lung cancer cell lines. The in vitro results showed that these nanoparticles are dual responsive and more potent in killing cancer cells than free drugs. In general, this manuscript is well-written and easy to follow. Therefore, I recommend a minor revision. There are a few suggestions.

1.       Line 141, “NPs were lyophilized for 2 days to collect DSNPs”. However, it appears that the NPs were lyophilized without any cryoprotectant. Please measure the changes in DLS size or TEM images before and after the lyophilization process.

2.       Line 160-162, the amount of NU7441 and Cisplatin was determined at 288 NM and 348 nm by a UV spectrophotometer. Please provide the full UV spectrum of NU7441 and cisplatin. There is a concern that these two drugs may have overlapped in UV spectrums leading to an inaccurate measurement.

3.       Figure 7 described the therapeutic efficacy of E-DSNP to AT1 and A549 cells, and suggested that E-DSNP binds poorly to AT1 cells. Please add references to explain the different expression levels of EphA2 in these two cell lines. If not, please provide western blot results or at least include the AT1 uptake result in figure 5.

Reviewer 2 Report

Comments

In the present manuscript entitled “Lung Cancer Targeted Chemoradiotherapy via Dual-Stimuli Responsive Biodegradable Core-Shell Nanoparticles,” the authors have made an excellent attempt to deliver ephrin functionalized dual stimuli-responsive nanoparticles loaded with cisplatin and NU7441 for the treatment of lung cancer. However, the authors missed critical analysis (statistical analysis) or critical comparison between the groups such as between control, Cisplatin, NU7441, and their formulations...etc.  Find my comments below.

1)    In the materials section, the authors purchased TEM grids for the morphology analysis of nanoparticles. However, nanoparticle morphology was analyzed by SEM, if the authors don’t want to use them, why did they purchase or mentioned them?

2)    Authors should include cell culture conditions and from where cells are procured.

3)    In the synthesis of DSNPs, what are GNP and HNP? If it is Glutathione and Hyaluronic acid nanoparticle, abbreviate if it comes the first time.

4)    In the drug release kinetics sections, the authors have missed critical comparative analysis between different groups. Authors used only E-DSNPs, why not other groups such as Cisplatin nanoparticles or NU7441 nanoparticles. At least the authors should compare drug releases between E-DSNPs and DSNPs. Moreover, data is represented for E-DSNPs (in figures 2 C &D) and explained for DSNPs (line 321) please maintain consistency throughout the manuscript.  How much PBS quantity was used for the drug release study?

5)    What is the PDI of the prepared nanoparticles.

6)    How freeze-drying is performed? with or without cryoprotectants? If with cryoprotectant what was the cryoprotectant and what ratio it is used? If without cryoprotectant, how do the authors confirm changes in particle size, drug content, and aggregation of particles? 

7)    How the Coumarin-6 was loaded into nanoparticles, whether loaded into the core or shell of the nanoparticles. How the loaded C-6 was quantified? Why the free C-6 group was not included in the study? Authors should compare the uptake study with free drug (C-6), nanoparticles, and functionalized nanoparticles.

8)    From the line 213-223 font size is different, correct it.

9)    Why did the authors perform the cytotoxic analysis of nanoparticles for 24 h, meanwhile cytotoxicity and efficacy studies were performed for drug-loaded nanoparticles for 72 h. if the authors want to check the safety of the nanoparticles in the healthy cell line, the treatment duration should be the same in both cases.

10) In the hemocompatibility study, the authors kept water as a positive control. However, in the results, no bar graphs were shown regarding the same. Furthermore, results were shown for E-DSNPs, while studies were performed for DSNPs.

Reviewer 3 Report

In this study, the author prepared a targeted nanomedicine with core-shell structure, which achieved significant effect on killing cancer cells. However, several issues should be addressed before acceptance.

1. The zeta potential of nanoparticles should be provided.

2. To observe the core-shell structure, the TME image should be taken.

3. It is better to use HPLC to measure the drug loading content.

4. There exist a lot of HAnase in the body and tumor, which also make degradation of HA, thus, it is necessary to employ radiation to trigger the drug release.

5. In drug release studies, without radiation, both two drugs were hardly released. However, in MTT assay, without radiation, why nanomedicine could still induce cytotoxicity on cancer cells?

6. Several related and latest references on HA-targeted nanomedicine should be cited and discussed, which were listed as following:

CHINESE CHEMICAL LETTERS, 2021, 32:1731-1736

CHINESE CHEMICAL LETTERS, 2021, 32:2117-2126

Nano-Micro Letters, 2021, 12:99.

Bioactive Materials, 2021, 6: 1513-1527

In this study, the author prepared a targeted nanomedicine with core-shell structure, and achieved significant effect on killing cancer cells. However, several issues should be addressed before acceptance.

1. The zeta potential of nanoparticles should be provided.

2. To observe the core-shell structure, the TME image should be taken.

3. It is better to use HPLC to measure the drug loading content.

4. There exist a lot of HAnase in the body and tumor, which also make degradation of HA, thus, it is necessary to employ radiation to trigger the drug release.

5. In drug release studies, without radiation, both two drugs were hardly released. However, in MTT assay, without radiation, why nanomedicine could still induce cytotoxicity on cancer cells?

6. Several related and latest references on HA-targeted nanomedicine should be cited and discussed, which were listed as following:

CHINESE CHEMICAL LETTERS, 2021, 32:1731-1736

CHINESE CHEMICAL LETTERS, 2021, 32:2117-2126

Nano-Micro Letters, 2021, 12:99.

Bioactive Materials, 2021, 6: 1513-1527

Reviewer 4 Report

The reviewer has few major concerns:

1.  line 118: This NP complex has  a GNP core. Doesn't day how they are made. In the table in Figure 2A, size of GNPs are written as 187 nm. This is pretty high core size for GNPs.  Is this the core size of GNPs?  Provide SEM images. 

2. Can you provide SEM images of final NPs? The Figure 2B shows a huge variation in sizes and need to provide a rationale for that.

3. Also provide Zeta potential data for your NP system at each step.  

4. What is "A" in figure caption. It is missing. Why there is a difference between cell cytoplasm visualization two sets of images in 'A".

5. The results in Figure 8 need further clarification. Error bars cannot be seen in figure 4 for three cases. This is unusual. These are biological experiments and there should be a visible error. I see some problems when i see clonogenic data in figure 8A. You should see at least 50 colonoes at the end of the time period to make a confident conclusion. For example, last dish in the right hand side, i see probably one colony. Therefore, i am not confident about the conclusions made. 

6." All results were displayed as mean ± SD, and quadruplet samples (n=4) 300 were used for each experiment if not specified.".  In clonogentc data, it should be repeat of 3 independent experiments and each data point should have at least three dishes.

6. Radiation is a critical part of this study. the energy of radiation and rest of the details are missing. 

7. the role of GNPs and its contribution is not clear.

8. NU7441 should be explained in the abstract. The rationale for using it is not given.

9. In addition to clonogenic assay, may a cell proliferation assay or DNA double strand breaks assay could have been useful to solidify your findings. The reason is you are using NU7441. NU7441 is an inhibitor of DNA-dependent protein kinase (DNA-PK).

Round 2

Reviewer 2 Report

Responses are satisfactory.

Author Response

Thank you very much for help improving the manuscript